# Enhancing of CO Uptake in Metal-Organic Frameworks by Linker Functionalization: A Multi-Scale Theoretical Study

**Charalampos G. Livas** [1] , **Emmanuel Tylianakis** [1,2] **and George E. Froudakis** [1,*]

1 Department of Chemistry, Voutes Campus, University of Crete, GR-71003 Heraklion, Crete, Greece; chemp1167@edu.chemistry.uoc.gr (C.G.L.); tilman@materials.uoc.gr (E.T.)

2 Department of Materials Science and Technology, Voutes Campus, University of Crete, GR-71003 Heraklion, Crete, Greece

\* Correspondence: frudakis@uoc.gr

**Abstract:** In the present work, the interaction strength of Carbon Monoxide (CO) with a set of forty-two, strategically selected, functionalized benzenes was calculated. Our ab initio calculations at the MP2/6-311++G** level of theory reveal that phenyl hydrogen sulfate ($-OSO_3H$) showed the highest interaction with CO ($-19.5$ kJ/mol), which was approximately three times stronger compared with the unfunctionalized benzene ($-5.3$ kJ/mol). Moreover, the three top-performing functional groups ($-OSO_3H$, $-OPO_3H_2$, $-SO_3H$) were selected to modify the organic linker of IRMOF-8 and test their ability to capture CO at 298 K for a wide pressure range. Our Grand Canonical Monte Carlo (GCMC) simulations showed a significant increase in the CO uptake in the functionalized MOFs compared with the parent IRMOF-8. It is distinctive that for the volumetric uptake, a $60\times$ increase was observed at 1 bar and $2\times$ was observed at 100 bar. The proposed functionalization strategy can be applied for improving the CO uptake performance not only in MOFs but also in various other porous materials.

**Keywords:** Carbon Monoxide storage; metal–organic frameworks (MOFs); multi-scale approach

## 1. Introduction

Carbon monoxide (CO) is a flammable, colorless, odorless, tasteless gas with a marginally lower density than air. It is naturally present in the Earth's atmosphere (approximately 80 ppb) with much of this build-up caused by human activities such as vehicles exhausts, and gas stoves. Natural events such as volcano eruptions and fires also produce significant quantities of CO [1]. CO is a problematic air pollutant that upon emission into the atmosphere has proven negative health effects and contributes to climate change. It has an indirect effect on radiative forcing by elevating concentrations of direct greenhouse gases, including methane and tropospheric ozone. It does this by reacting with other atmospheric constituents that can reduce methane, and it is also oxidized to carbon dioxide and ozone through natural processes in the atmosphere [2].

A high concentration of Carbon Monoxide in the air reduces the amount of oxygen that can be transported into the bloodstream and to critical organs such as the brain and heart. Within enclosed environments, a higher concentration of CO may lead to more serious health issues such as dizziness, unconsciousness, and death. Every year, 170 people in the United States die from Carbon Monoxide poisoning [3].

In light of the health and environmental concerns surrounding Carbon Monoxide pollution, it is an important reagent within the chemical engineering industry. Syngas, which is created by partially oxidizing hydrocarbon feedstocks, is a mixture of $H_2$ and CO and constitutes the main source of CO. The use of CO as a reagent usually requires a purity of 99 mol % [4,5]. In this context, a combination of cryogenic distillation, acid gas removal, and dehydration must be applied. Accomplishing high purity induces additional costs and separation steps. In general, separation methods are energy-intensive processes, contributing 30–40% of the capital and operating costs of the chemical industry

and representing approximately 10–15% of global energy consumption [5]. From a cost and sustainability standpoint, a modest technical advancement can have a big impact.

Among the solutions to CO purification investigated is the discovery of new materials with high selectivity toward CO in a mixture of gases and high adsorption loading. Metal–organic frameworks (MOFs) have attracted wide interest as promising candidates for gas adsorption applications. Introduced by O. Yaghi in 1995 [6], MOFs are organic–inorganic hybrid materials composed of metal ions or clusters interconnected through organic linkers. The structure of most MOFs is characterized by an open porous framework resulting in important properties such as low density, high surface area and high porosity. These attributes lead to the adsorption of gases in more manageable volumes at relatively low pressures. Due to the variety and tunability of their structure, MOF properties can be optimally tuned for each specific application. There exist numerous potential applications in gas storage, ion exchange, molecular separation and heterogeneous catalysis [7–12].

Recently, important strategies have been employed to enhance the Carbon Monoxide storage capacity of MOFs. Introducing open metal sites in the metal oxide part, functionalizing the organic linker and controlling the pore size (e.g., increase in BET surface area, impregnation) constitute some of them [13]. Karra and Walton performed atomistic Grand Canonical Monte Carlo (GCMC) simulations and calculated the adsorption of CO in four MOFs; IRMOF-1, IRMOF-3, $Zn_2[bdc]_2[dabco]$, and Cu-BTC [14]. The latter performed better through the whole pressure range and reached an adsorption capacity of 11 mmol/g at 45 bar and 298 K. Bloch et al. synthesized six metal–organic frameworks of the M2(dobdc) (M = Mg, Mn, Fe, Co, Ni, Zn; dobdc4− = 2,5-dioxido-1,4-benzenedicarboxylate) structure type [15]. Adsorption isotherms indicated reversible adsorption and loadings as high as 6.0 mmol/g. Deng et al. synthesized 18 multivariate MOF-5 type structures that contained up to eight distinct functionalities such as $-NH_2$ −Br and $-Cl_2$ [16]. The functionalization of the organic linker led to an increase in volumetric uptake of up to 80% at 298 K and 1 bar.

In this work, we are aiming to propose new MOF materials with enhanced CO adsorption capacity by introducing strategically selected Functional Groups (FGs) into the organic linkers of the MOFs for enhancing the interaction with CO, resulting in an increase in the adsorption capacity.

Previous studies of our group focusing on different gases have shown that this approach generates remarkable advantages over their parent frameworks [17–20]. For example, theoretical studies by Frysali et al. [17] showed that the introduction of sulfate anion in the phenyl ring enhances the interaction with $CO_2$ (−22.6 kJ/mol) by almost two times. Moreover, the work of Giappa et al. [18] showed that the introduction of several FGs such as hydroxyl sulfate ($-OSO_3H$) can enhance the interaction with hydrogen by up to 80%. Moreover, taking into account FGs that have been already studied for hydrogen adsorption provides the ability to explore a potential application of syngas separation.

Since many organic linkers used in MOFs consist of aromatic rings, such as benzene and naphthalene, a substituted benzene was chosen as a functionalization platform for our computationally demanding ab initio calculations. A set of 42 strategically selected, functional groups were selected by chemical intuition and findings from previous studies [17–20]. In order to identify the most promising functional groups, ab initio calculations were conducted, and the binding energy of CO to 42 functionalized phenyl rings was calculated. To understand the nature of the optimized systems, plots of the electrostatic potential of the functionalized benzenes and electron density redistribution plots of the dimers were generated.

In order to verify the effectiveness of the functionalization on the enhancement of CO uptake in MOFs, IRMOF-8 was selected as the platform, and its organic linkers were functionalized with the 3 top performing FGs. The gravimetric and volumetric uptake was obtained at ambient temperature of 298 K and pressure up to 100 bar by performing Grand Canonical Monte Carlo (GCMC) simulations. Our results can serve as a general guide, as they can be applied to MOFs and numerous other porous materials.

## 2. Methodology

Ab initio calculations were employed to estimate the binding strength of CO with a large set of 42 FGs at the MP2/6-311++G** level of theory. Full geometry optimization was performed for all different conformations with the ORCA program package [21,22]. In addition, the binding energies were corrected for the Basis Set Superposition Error (BSSE) with the Counterpoise (CP) method as proposed by Boys and Bernardi [23], since BSSE correction is important for nonbonding interactions.

Electrostatic potential maps of the functionalized benzene monomers and electron density redistribution plots of the dimers were constructed for understanding the nature of the interaction of carbon monoxide with the functionalized benzenes.

To determine how the functionalization of the organic linker impacts the adsorption capacities of CO in MOFs, we chose IRMOF-8 as a platform, and we modified all the organic linkers with the three top performing functional groups. Then, Monte Carlo simulations in the Grand Canonical ensemble were carried out at 298 K for a pressure range up to 100 bar using the RASPA software package [24].

We used the Lennard–Jones $6-12$ (LJ) [25] + Coulomb potentials to describe the interactions between the host and the sorbate atoms. Each atom of the host or the guest was treated explicitly. The following formula constitutes the potential used for this research:

$$V_{ij} = 4\varepsilon_{ij}\left[\left(\frac{\sigma_{ij}}{r_{ij}}\right)^{12} - \left(\frac{\sigma_{ij}}{r_{ij}}\right)^{6}\right] + \frac{q_i q_j}{4\pi\varepsilon_0 r_{ij}}$$

$\varepsilon_0$ is the vacuum permittivity constant, $q_i$ and $q_j$ are the corresponding partial charges for atoms $i$ and $j$, $r_{ij}$ is the interatomic distance between interacting atoms $i$ and $j$, and $\varepsilon_{ij}$ and $\sigma_{ij}$ are the LJ potential well depth and the repulsion distance between atoms $i$ and $j$, respectively. Carbon Monoxide and IRMOF-8 were considered to be rigid, and we represented them by atomistic models. The first, Carbon Monoxide, was treated using the TraPPE [26] model, and during the simulations, the bond length was 1.137 Å and was not allowed to vary. For the electrostatic interactions, the carbon atom of the CO molecule had a point charge of 0.0223 e, and the oxygen atom had the opposite charge, i.e., $-0.0223$ e. We used potential parameters according to the TraPPE model developed by Potoff and Siepmann for the van der Waals interactions, i.e., $\varepsilon = 37.15$ K and $\sigma = 3.55$ Å for the carbon atom and $\varepsilon = 61.57$ K and $\sigma = 2.95$ Å for the oxygen. For each MOF structure, the necessary potential parameters were taken from the UFF force field [27]. Lorentz–Berthelot mixing rules were used to describe the CO−IRMOF-8 interactions. These parameters are proven to describe correctly the dispersion interactions in previous studies [28,29], but the applicability in the current system was also evaluated by ab initio calculations. In order to do that, we performed ab initio calculations at the MP2/6-311++G** level of theory and calculated the potential energy surface of the CO approaching the functionalized benzene. Then, we fitted the UFF parameters to reproduce the quantum chemical data. The final parameters are available from the authors upon request. Details can be found in the Supporting Information. The cutoff value for the LJ potential was set to 12.8 Å. Electrostatic interactions were also considered for the interaction of CO with the host material, and they were treated with the Ewald summation method. We calculated the partial charges of the framework atoms as implemented in Orca with the CHELPG method. The resulting partial charges were slightly changed in order to keep the molecular structure in the periodic box neutral.

## 3. Results and Discussion

In this work, the CO interaction with 42 functionalized benzene molecules was theoretically investigated. In Figures 1 and 2, the geometries and the corresponding binding energies of the top 10 energetically most favorable dimers of CO with the functionalized aromatic molecules can be seen. Details for the rest of the 42 functionalized benzene molecules can be found in Figures S1 and S2 of the supporting information. All optimized structures

are available upon request. From Figure 1, it can be seen that the optimized structures can be divided into two categories according to the CO position, that is, CO located straight above the benzene ring or toward the functional group. The binding energies calculated at the MP2/6-311++G** level, reported in Figure 2, range from 8.3 to 19.4 kJ/mol for CO–$C_6H_5CH_2NH_2$ and CO–$C_6H_5OSO_3H$, respectively. For comparison, the least favorable structure as seen in Figure S1 was CO–$C_6H_5SO_2Cl$ with a binding energy of 4.1 kJ/mol, and the binding energy with the unfunctionalized benzene ring was 5.4 kJ/mol.

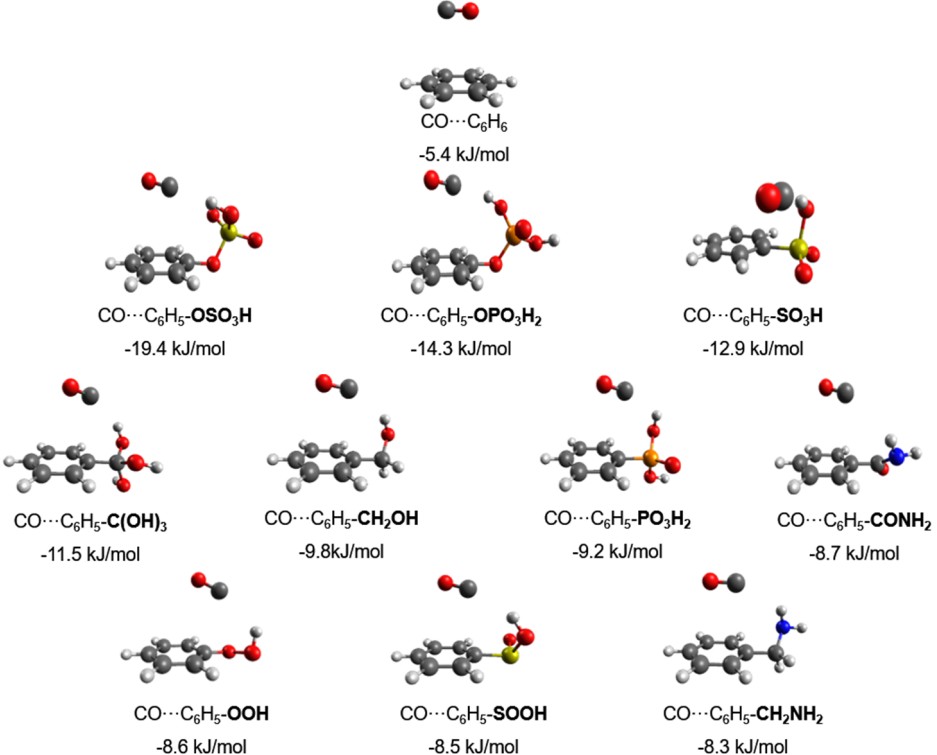

**Figure 1.** MP2/6-311++G** optimized geometries for the 10 best-performing FGs.

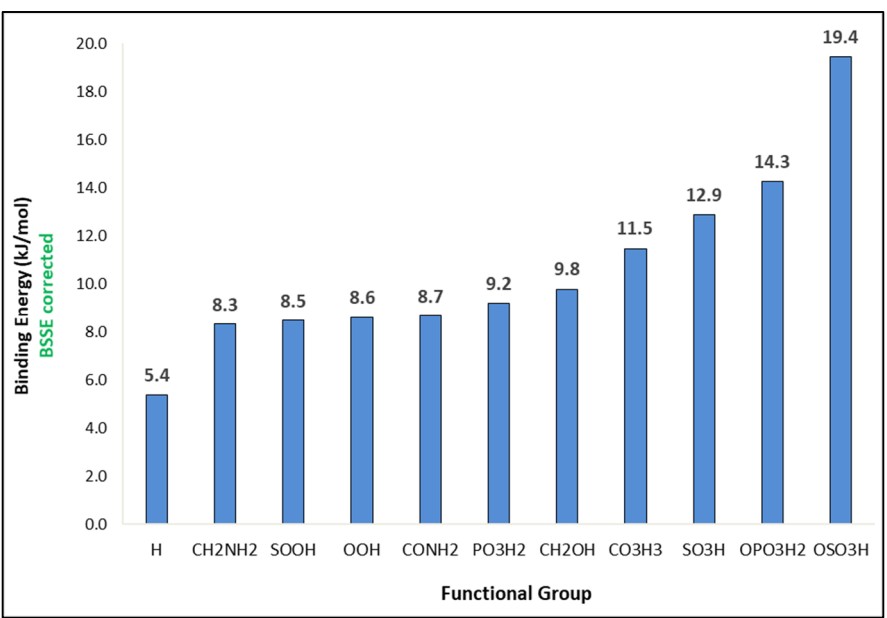

**Figure 2.** The 10 substituted benzenes with the highest CO binding energy and the unsubstituted benzene for comparison, calculated at the MP2/6-311++G** level of theory and corrected for BSSE.

The strongest interaction energy was found for $CO–C_6H_5OSO_3H$ (19.4 kJ/mol). In this structure, CO is located above the benzene ring and close to the functional group (Figure 1). The complex is stabilized by the interaction of the carbon atom of CO with the hydrogen atom of the functional group. The distance between the carbon and the hydrogen atom is 2.071 Å, and the bond length of CO is 1.138 Å, which is the smallest observed in all 42 optimized structures. In addition, the dimer is further stabilized by the $\pi$–$\pi$ interaction between the carbon monoxide molecule and the substituted benzene. In this way, CO takes advantage of both the $\pi$ system of the ring and the FG's influence.

The next two energetically more favorable functional groups, $OPO_3H_2$ and $SO_3H$, have binding energies with CO of 14.3 kJ/mol and 12.9 kJ/mol, respectively. We can see that the CO molecule is again approaching a hydrogen atom of the functional group. The distances between the Carbon atom of the CO and the Hydrogen atom of the FG are 2.160 Å for $OPO_3H_2$ and 2.173 Å for $SO_3H$. Concerning the other functional groups, we observe that the distance between the C and the H atoms increases as the binding energy decreases. In almost all of the 20 top performing functional groups, we can see this correlation diagram between the interaction energy and the change in the C–H distance (Figure 3). It can be clearly seen that as the interaction of CO increases, an almost linear decrease in the distance mentioned before is observed.

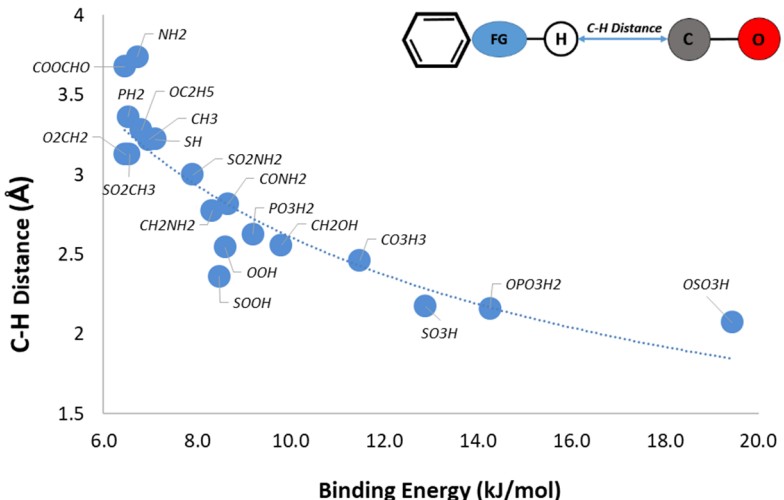

**Figure 3.** Plot of the distance between the Carbon atom of the CO molecule and the Hydrogen atom of the functional group as a function of the binding energy.

In the optimized geometries found for CO with the functionalized benzenes, we observed that the CO is oriented in a parallel mode above the plane of the benzene ring, with an exception in the case of $C_6H_5SO_3H$. In all cases except the latter, CO is displaced from the center of mass of the benzene ring, which indicates that there is an additional interaction between the oxygen atom of the CO and the atoms of the polar groups on top of the expected interaction between the quadrupole moment of CO and the $\pi$ system of benzene. This phenomenon has also been mentioned in previous studies on the interaction of $CO_2$ with molecular systems that contain aromatic rings [17].

Electrostatic potential maps and electron density redistribution plots can qualitatively explain the aforementioned results, i.e., the optimized geometries and the corresponding binding energies. The charge distribution of a molecule can be visualized in electrostatic potential maps. Electrostatic potential maps can be useful in explaining and predicting the geometry of the molecular system and the nature of the interactions between two molecules, especially when electrostatic interactions are important in describing their interaction.

The electrostatic potential maps of the CO and the top 10 performing functionalized benzenes are presented in Figure 4. Figure S3 of the supporting information shows the electrostatic potential maps for all the other aromatic molecules. The electrostatic potential map of CO is separated into two electron-rich and one electron-poor region. We observe

areas of enhanced electrostatic potential in the two atoms and a more neutral part in the middle of the molecule. We also see that around the carbon atom, we have a higher density of electrons than around the oxygen atom. This, in combination with the low density observed in the hydrogen of the functional groups, plays an important role in the orientation of CO in the dimer.

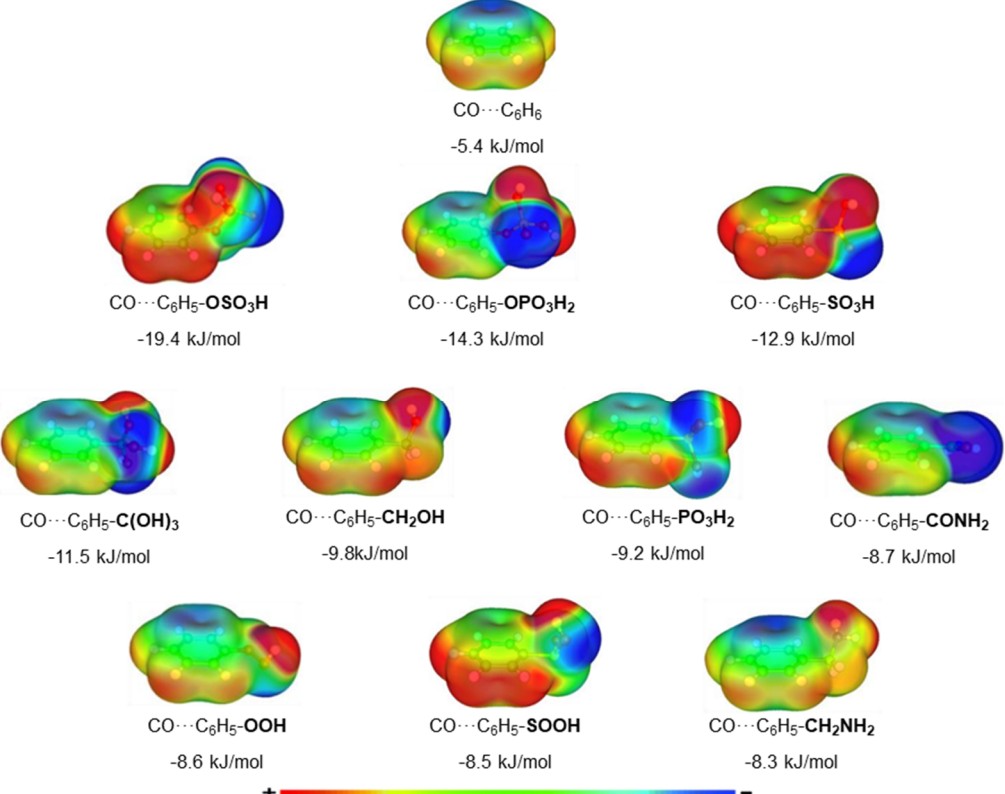

**Figure 4.** Electrostatic potential maps of the strongest interacting $C_6H_5$-X monomers. Calculated using the MP2/6-311++G ** method with ORCA 4.2 and visualized with gOpenMol [30,31] The varying intensities are ranging from −0.03 to +0.03 Hartree·$e^{-1}$. Red: electron-poor regions—high potential value, Blue: electron-rich regions— low potential.

Electron-density redistribution plots for the top 10 complexes and the unfunctionalized benzene ring are presented in Figure 5. Figure S4 of supporting information shows the density redistribution plots for all the other aromatic molecules interacting with CO. It is evident that the largest redistribution of electron density is observed for the CO–$C_6H_5OSO_3H$ complex, whereas CO–$C_6H_6$ presents the lowest redistribution. It can be observed that for complexes with the highest binding energies, there is an electron density gain (blue regions) on the C atom of CO and a relative electron density loss (green regions) on the O atom of CO. This is very well visualized in the case of the interaction of CO with the $OPO_3H_2$ functionalized benzene. Additionally, the existence of an acidic hydrogen atom in the FG induces an extra stabilizing effect on the complex due to the shift of electron density between that Hydrogen atom and the neighboring Carbon atom of the CO. Near H atoms, an electron density loss is detected, while an electron density gain is recorded surrounding the CO's nearest C atom. This effect can be clearly seen in the interaction of CO with -$OSO_3H$, -$OPO_3H_2$, and -$SO_3H$ functionalized benzenes.

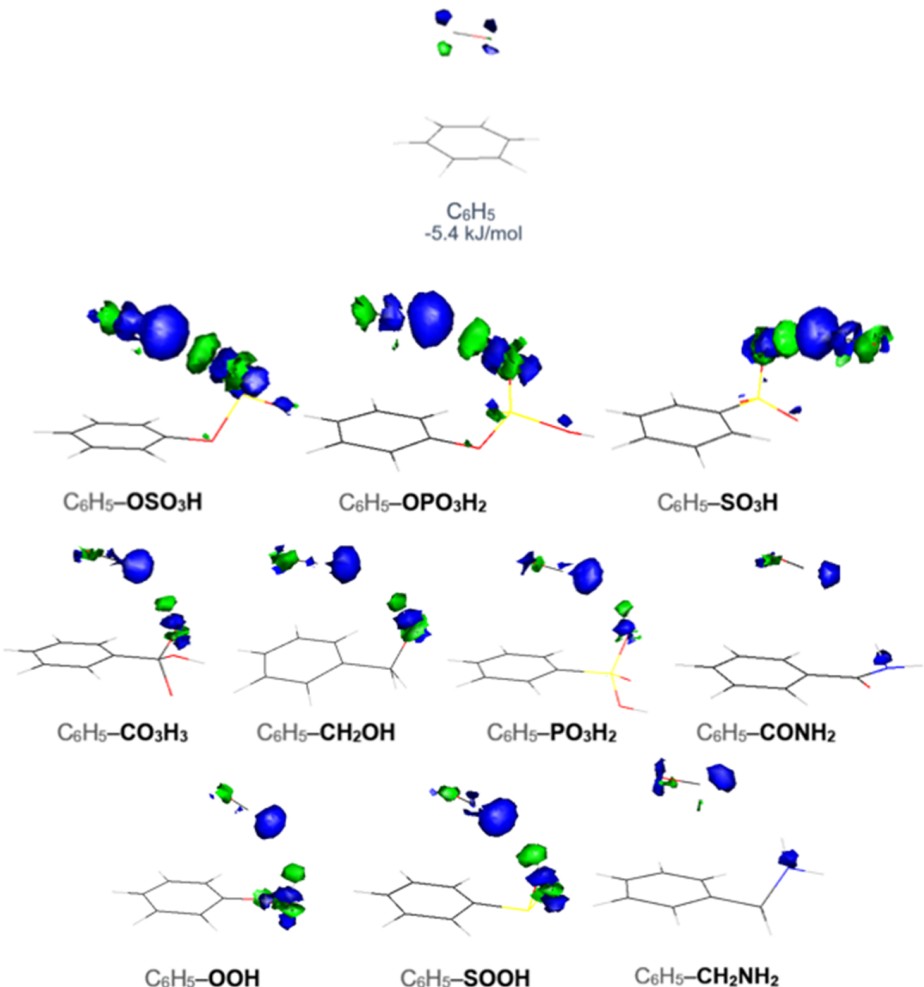

**Figure 5.** Electron-density redistribution plots of the optimized geometries of the strongest interacting $CO \cdots C_6H_5$-X complexes. Blue and green regions that represent areas that gain or lose electron density, respectively.

In order to verify the effectiveness of the functionalization on the enhancement of CO uptake in MOFs, IRMOF-8 was selected as the platform and functionalized with the 3 top performing functional groups. The gravimetric and volumetric uptakes were obtained at an ambient temperature of 298 K and pressure up to 100 bar by performing Grand Canonical Monte Carlo (GCMC) simulations. In Figures 6 and 7, the gravimetric and volumetric uptake isotherms for the functionalized IRMOF-8 structures with the three top-performing functional groups ($-OSO_3H$, $-OPO_3H_2$, $-SO_3H$) are shown together with the one of the parent materials for comparison. From both gravimetric and volumetric isotherms, the enhancement of the uptake due to functionalization is clear in all pressure ranges. The improved interaction of these functional groups with CO is clearly reflected in the corresponding gas adsorption of the modified MOFs. As demonstrated in Figure 7b, the improved performance of the modified structures is more obvious at the low loading limit, i.e., in the low pressure range. This is to be expected because at this limit, the interaction energy dominates over other parameters such as pore volume and surface area in determining adsorption.

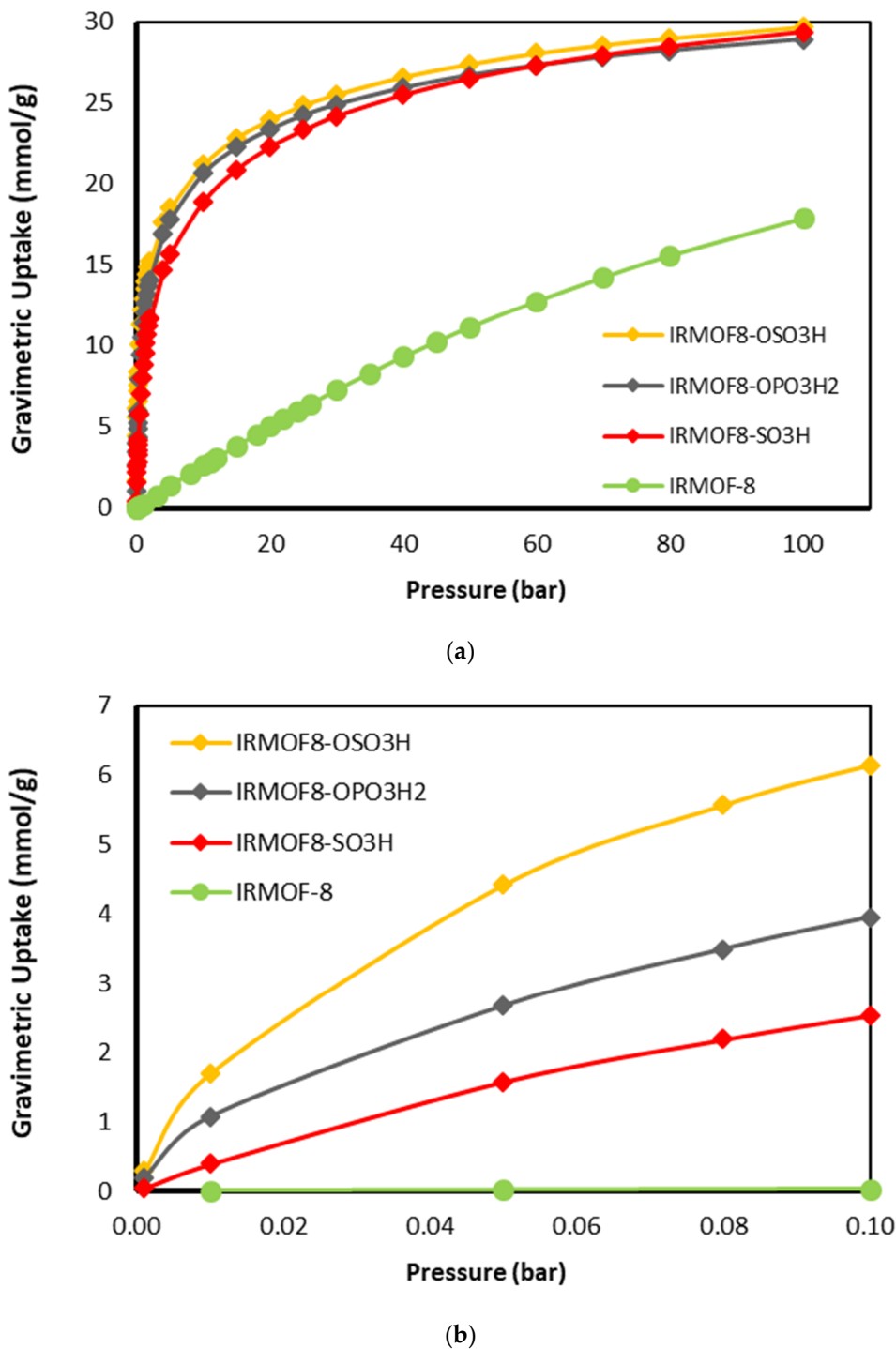

**Figure 6.** Absolute gravimetric isotherms for IRMOF8 and IRMOF8-n (n: -OSO$_3$H, -OPO$_3$H$_2$, -SO$_3$H) at T = 298 K and pressure ranges up to (**a**) 100 bar and (**b**) 0.10 bar.

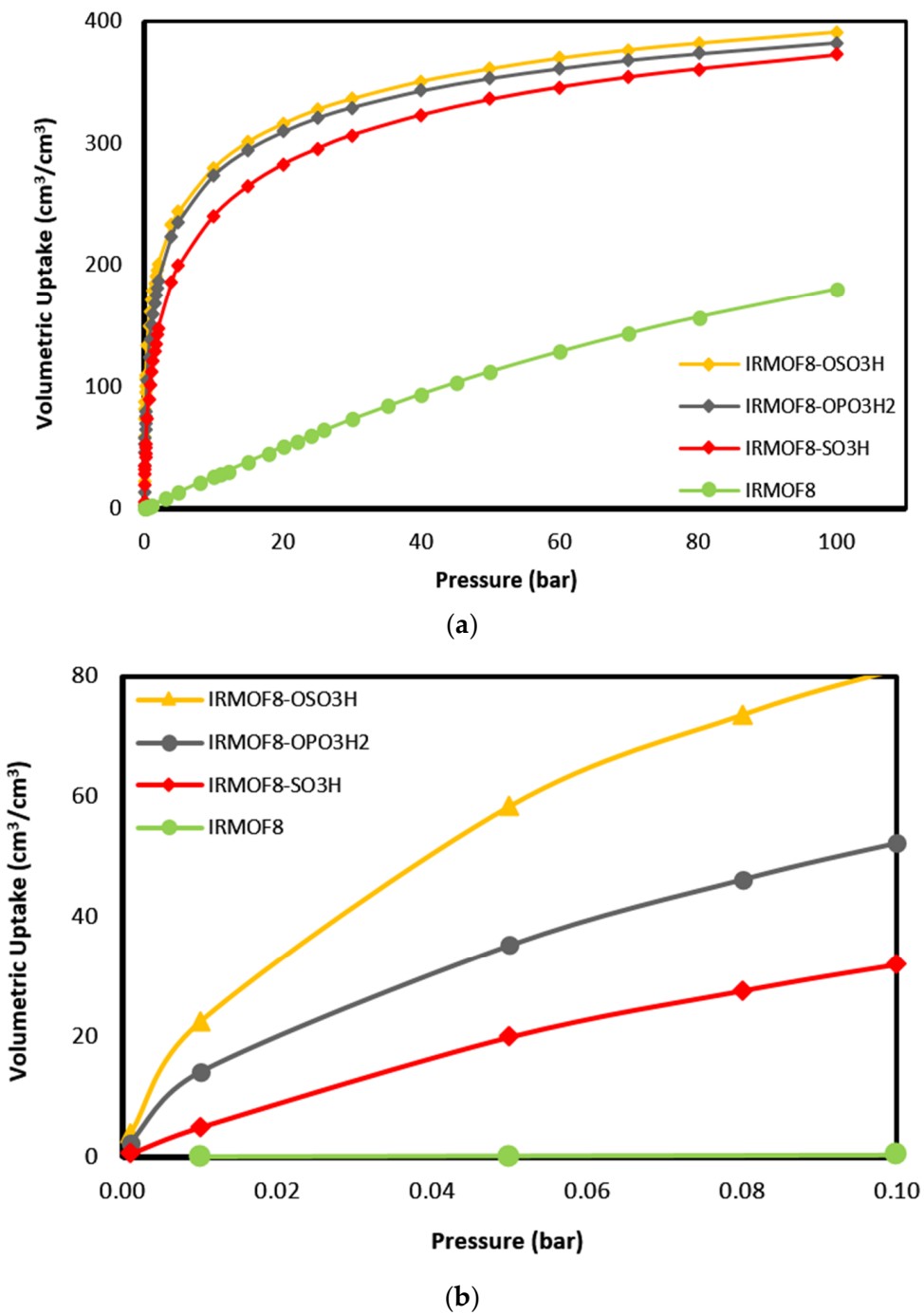

**Figure 7.** Absolute volumetric isotherms for IRMOF8 and IRMOF8-n (n: -OSO$_3$H, -OPO$_3$H$_2$, -SO$_3$H) at T = 298 K and pressure ranges up to (**a**) 100 bar and (**b**) 0.10 bar.

This is further supported by snapshots obtained during the GCMC simulations at 0.6 bar and 298 K for the -OSO$_3$H modified MOF and unmodified IR-MOF8 (Figure 8). In perfect agreement with the corresponding isotherms, the functionalized MOF hosts considerably more CO molecules than the parent structure due to the stronger binding sites introduced to the framework by functionalization. This is also verified by the fact that the CO molecules are located closer to the functional groups at low loading conditions.

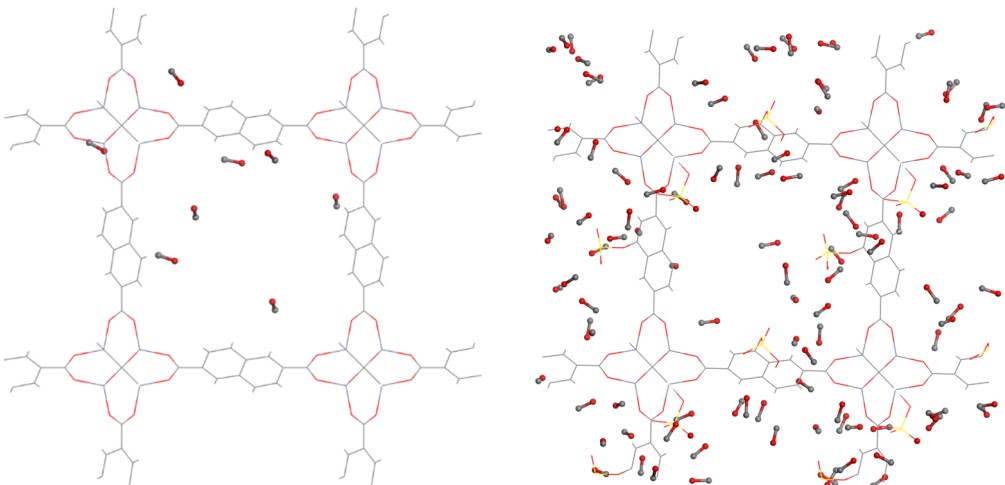

**Figure 8.** Snapshots for (**left**) IRMOF8 and (**right**) IRMOF8-OSO$_3$H at T = 298 K, *p* = 0.60 bar.

## 4. Conclusions

In summary, we studied how the functionalization of the organic linker impacts the carbon monoxide adsorption in MOFs. Ab initio calculations were performed to study the interaction energies of carbon monoxide with 42 substituted benzenes. MP2/6-311++G** calculations showed that the highest interaction energy (−19.5 kJ/mol) with CO was found for phenyl hydrogen sulfate (-OSO3H), which is approximately three times stronger than the corresponding binding energy for benzene (−5.3 kJ/mol). Qualitatively, the difference in the binding energies of substituent benzenes can be explained by the electrostatic potential maps and the electron density redistribution plots. When this redistribution of the carbon atom of CO is more intense, higher binding energy was found. In most of the studied cases, the CO molecule preferred a position on top of the ring and near the functional groups. This geometrical configuration manages to maximize both the interaction between the aromatic ring and the CO molecule and the interaction between the acidic hydrogen atom of the functional group and the CO's carbon atom. In addition, it is worth noting that substituents that are electron acceptors (F, CN) have weaker binding energies with CO, whereas substituents that are electron donors (OH, CH$_3$, OC$_2$H$_5$) have stronger binding energies, which reinforces the impact of hydrogen bonding. Moreover, we functionalized IRMOF-8 with the three top-performing functional groups (-OSO$_3$H, -OPO$_3$H$_2$, -SO$_3$H) and employed Grand Canonical Monte Carlo simulations at ambient temperature and up to 100 bar. All modified MOFs showed a 46× increase at 1 bar and a 40% increase at 100 bar in gravimetric adsorption. Similar results were seen for the volumetric uptake, with a 60× and 113% increase at 1 and 100 bar, respectively. The obtained isotherms dictate that our functionalized MOFs can be further studied for applications in environments with high CO emissions such as vehicle exhausts. Furthermore, taking into consideration the results of a previous study by Giappa et al. [18], we can conclude that H$_2$ shows two to three times less binding energy with C$_6$H$_5$-n (n= -OSO$_3$H, -OPO$_3$H$_2$, and -SO$_3$H) compared with a CO molecule. Resultantly, the three modified IR-MOF8 can potentially be used also for separating syngas. We believe that the results obtained from this study can both serve as high accuracy reference and guide synthetic experiments toward materials with high CO storage capacity and selectivity.

**Supplementary Materials:** The following supporting information can be downloaded at: https://www.mdpi.com/article/10.3390/chemistry4020043/s1, Figure S1: Sorted binding energies (kJ/mol) of the CO···C6H5-X systems under study, calculated at the MP2/6-311++G** level of theory. All interaction energy values have been corrected for the Basis Set Superposition Error (BSSE) by the full counterpoise method; Figure S2: Global minima geometries of all the systems in this study; Figure S3: Electrostatic potential maps of the strongest interacting $C_6H_5$-X monomers. Calculated using the MP2/6-311++G** method with ORCA 4.2 and visualized with gOpenMol. The varying intensities are ranging from −0.03 to +0.03 Hartree·e-1. Red: Electron-poor regions—high potential value, Blue: electron-rich regions—low potential; Figure S4: Electron-density redistribution plots of the optimized geometries of the CO···$C_6H_5$-X complexes. Blue and green indicate the regions that gain and lose electron density upon the formation of the complex, respectively; Figure S5: Fitting of the ($\varepsilon$, $\sigma$) parameters of the UFF potential on the QM data obtained from the ab initio scan of CO over benzene; Figure S6: Fitting of the ($\varepsilon$, $\sigma$) parameters of the UFF potential on the QM data obtained from the ab initio scan of CO over (a) $C_6H_5$-$OSO_3H$ (b) $C_6H_5$-$OPO_3H_2$ (c) $C_6H_5$-$SO_3H$; Figure S7: Gravimetric (a) and volumetric (b) Carbon Monoxide uptake isotherms for IRMOF-8 and IRMOF-8-n (n: -$OSO_3H$, -$OPO_3H_2$, -$SO_3H$) at T = 298 K [21–23,27,30–33].

**Author Contributions:** Conceptualization, C.G.L. and G.E.F.; methodology, C.G.L., E.T. and G.E.F.; software, C.G.L. and E.T.; validation, C.G.L. and E.T.; formal analysis, C.G.L.; investigation, C.G.L.; resources, G.E.F.; data curation, C.G.L.; writing—original draft preparation, C.G.L.; writing—review and editing, C.G.L. and G.E.F.; visualization, C.G.L.; supervision, G.E.F. All authors have read and agreed to the published version of the manuscript.

**Funding:** This research received no external funding.

**Institutional Review Board Statement:** Not applicable.

**Informed Consent Statement:** Not applicable.

**Data Availability Statement:** The data presented in this study are available in Supplementary Materials.

**Conflicts of Interest:** The authors declare no conflict of interest.

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
