# Peer review of "Enhancing of CO Uptake in Metal-Organic Frameworks by Linker Functionalization: A Multi-Scale Theoretical Study"

_chemistry, doi:10.3390/chemistry4020043_

Round 1

Reviewer 1 Report

Froudakis and coworkers present a neat, well-conceptualized study examining the binding of CO to the ca. 40 possible functionalized phenyl linkers using MP2, followed by GCMC calculations on their leading linkers.

I have only a few small comments to make:
How were the initial geometries chosen? And how many per C6H5-FG?

In the beginning of their results and discussion section (Fig 1 and 2), the authors state that there are two clear groups of structures - I think they could make those groups a bit clearer in Fig 1 and maybe even colour code Fig 2 accordingly. It would also be nice if the BEs were added to Fig 1 like it already is for the base benzene linker.

In future studies (I think it would be too expensive to do now), but it would also be interesting to do GCMC of one of the worst performing linkers as a kind of negative test - it would be useful in the same way that the authors use Giappa et al's results here.

Could the authors also provide (as a zipfile or similar) xyz files of their optimized geometries?

Minor comments:
Line 76: resulting in an increase in the adsorption capacity.
In many places the author's spellcheck has been not nice and changed "dimers" to "dimmers", the authors obviously need to change that back and perhaps make an appeasement offering to the Gods of Spellcheck.
Line 235, I think "lose" electron density is better than "miss" electron density
Fig 8: Can a figure without the black background work? If not, I understand.
Line 290: enforces -> reinforces

Reviewer 2 Report

C.G. Livas and coworkers presented a theoretical study on the analysis and materials design of IRMOF-8 derivatives with improved CO uptake. This research is very relevant in the context of the environmental concerns of CO pollutant, as well as finding cost-effective methods to obtain high purity CO. In this work 42 Functional Groups attached to the benzene ring are considered and the CO binding energy evaluated using high level MP2/6-311++g** method including counterpoise correction. From the best 3 candidates, the CO adsorption is computed using Gran Canonical Monte Carlo simulations with parametrized UFF interaction potentials obtained from quantum chemical computations. This work is very well presented including the context, the methods, the results as well as the supporting information, and I recommend publication.

The only minor comment is few references are missing. There are few statements not supported by literature for which a reference should be added:

Page 2: “Recently, important strategies have been employed to enhance the carbon monoxide storage capacity of MOFs. Introducing open metal sites in the metal oxide part, functionalizing the organic linker and controlling the pore size (e.g. increase of BET surface area, impregnation) constitute some of them.”

Page 5:” This phenomenon has also been mentioned in previous studies on the interaction of CO2 with molecular systems that contain aromatic rings.”

And to my knowledge in English “dimmer” is not correct to refer to a “dimer”

Reviewer 3 Report

In this paper, Livas et al. presented a comprehensively theoretical study on the functionalization of IRMOF-8 to enhance its CO uptaking feature. The simulation clearly suggests that the functionalization of -OSO3H in IRMOF-8, agrees with the enhanced binding energy calculation. The simulation is well performed with key references cited properly.

I believe the selective adsorption between CO and CO2 will also be a very interesting project. I hope the authors could make a follow-up effort in this direction in the future.

Here are some points that need to be addressed before publication,

1.      In the y-axis of gravimetric isotherms, the unit presented in mmol/gr is less common for experimental researchers. I suggest using mmol/g instead.

2.      The schematic representation shown in Figure 3 is misleading. I suggest that the hydrogen belonging to the functional group should be presented in the schematic representation.

3.      The electrostatic potential shown in Figure 4 is not an ideal method to analyze the interactions between FGs and CO. I suggest exhibiting their dipole moment correspondingly. I believe dipole-dipole interaction, in this case, is critical for the binding of CO to FGs.
